# Efficacy of Segmentation for Hyperspectral Target Detection

**DOI:** 10.3390/s25010272

**Published:** 2025-01-06

**Authors:** Yoram Furth, Stanley R. Rotman

**Affiliations:** Department of Electrical and Computer Engineering, Ben-Gurion University of the Negev, Beer Sheva blvd 1, Beer-Sheva 84105, Israel; srotman@bgu.ac.il

**Keywords:** hyperspectral image, point target detection, Segmented Matched Filter, segmentation

## Abstract

Algorithms for detecting point targets in hyperspectral imaging commonly employ the spectral inverse covariance matrix to whiten inherent image noise. Since data cubes often lack stationarity, segmentation appears to be an attractive preprocessing operation. Surprisingly, the literature reports both successful and unsuccessful segmentation cases, with no clear explanations for these divergent outcomes. This paper elucidates the conditions under which segmentation might improve detector performance. Focusing on a representative algorithm and assuming a target additive model, the study examines all influential factors through theoretical analysis and extensive simulations. The findings offer fundamental insights and practical guidelines for characterizing segmented datasets, enabling a thorough evaluation of segmentation’s utility for detector performance. They outline the range of target scenarios and parameters where segmentation may prove beneficial and help assess the potential impact of proposed segmentation strategies on detection outcomes.

## 1. Introduction

Assuming the presence of a subpixel target within a hyperspectral image, traditional algorithms frequently fail in detection despite matching the target spectral signature. These algorithms typically employ the inverse covariance matrix for noise reduction. Our inquiry is directed towards elucidating the specific conditions under which using image segmentation can enhance the detector performance.

Historically, there have been conflicting reports on how beneficial segmentation is for detection. For instance, the generally positive results of [1,2,3] contrast with the negative findings reported by Pieper et al. [4]. These inconsistencies raised questions about the true value of segmentation. However, recent studies [5,6,7] have demonstrated that segmentation preprocessing can significantly improve detection performance, emphasizing that its success often depends on specific methods and conditions. These findings, combined with the earlier conflicting results, highlight the need for a deeper understanding of the factors that influence segmentation’s efficacy.

Focusing on the Normalized Segmented Matched Filter (NSMF) with a target additive model, this study explores segmentation’s role from foundational concepts to practical applications. It determines the conditions under which segmentation is likely to enhance detection efficacy for a given algorithm and data properties. We demonstrate how various parameters—such as the target’s strength, spectral signature, and the acceptable false alarm rate—impact segmentation’s contribution. Building on these insights, we aim to establish a predictive framework that outlines, for a given background, in which range of parameters segmentation is likely to improve detection.

In a previous study, we introduced the effect of the target’s spectral signature on segmentation efficacy [8]. Here, we extend this analysis to consider all relevant factors influencing segmentation’s efficacy in detection. We propose that the observed lack of significant improvement in some cases arises from intrinsic data properties and other detector-related factors. This study provides practical guidelines to predict what possible benefit can be gained if the detector uses the segmented data.

The introductory section outlines the foundational principles. Section 2 identifies key rules that explain how each factor simultaneously influences segmentation efficacy. Section 3 presents simulations and various tests on a standard dataset. Finally, Section 4 presents practical conclusions regarding the conditions under which segmentation improves detection efficacy.

### 1.1. Matched-Filter Target Detection

A standard algorithm for detecting small targets is the Matched Filter (MF), chosen in this study as the representative method for evaluating segmentation efficacy due to its widespread use and reliance on covariance-based whitening. Other algorithms, such as the Adaptive Cosine Estimator (ACE), also depend on covariance properties, suggesting that the findings here will generalize to ACE and similar methods. Ongoing research is testing this hypothesis.

Assuming the background data fit a multivariate normal distribution (MVN) once the background power has been mitigated and assuming an additive target model, it is essentially defined by $t^{T}\Phi^{-1}\left(x-m\right)$. For each pixel vector x, it subtracts the background m, whitens the residual by the inverse covariance matrix Φ, and matches the result to the desired target signature t. Using the Generalized Likelihood Ratio Test (GLRT), m and Φ are estimated from the data itself [9]. This provides a score to each pixel, related to the probability it contains a target. The target is then detected by testing the hypothesis that x contains only “background” against the hypothesis that it contains a target. In this study, we assume an additive target model of x≔x +pt, where p is the target power.

Consider the hypothesis that there is no target present in the dataset. While collecting statistics of such scores “without target” being present can be conducted directly on the original dataset (due to the overwhelming number of no-target pixels), robust evaluation for the “with target” hypothesis becomes possible using an implantation procedure, as explained in [10]. This procedure involves implanting desirable targets alternately into the dataset, enabling systematic analysis of algorithm performance across a range of controlled target-to-background interactions.

The distributions of scores for the two hypotheses (“with target” and “without target”) typically overlap, making it challenging to define an optimal detection threshold that balances detection rates with false alarms. The ROC curve is frequently used to optimize this threshold, as it provides a graphical representation of the trade-off. Additionally, the ROC curve serves as a metric of algorithm quality, with higher curves indicating greater cumulative detections for a given false alarm rate.

These two domains derive two different thresholds’ notations used in this paper: (i) “η”, indicating the *decision threshold* in the distributions’ domain, versus (ii) “th”, indicating the number of false alarms in the integral domain, called the *error threshold*. The relationship between these thresholds is:(1)$$th=\int_{\eta }^{\infty }f_{N}\left(z\right)dz,\;$$
where $f_{N}\left(z\right)$ is the scores’ distribution with no targets (N) present.

### 1.2. Normalized SMF

Since data are often not stationary and the variance varies throughout the data, estimating one global covariance from the whole image might be wrong [1]. On the other hand, calculating it locally is problematic due to insufficient statistics [11]. A common approach to address this challenge is segmentation: dividing the image into distinct areas based on the similarity of clustered points, estimating the covariance for each area individually, and applying the detector locally using the corresponding covariance. This results in two variations of the MF algorithm: a local version and a global version [12]:(2)$$SMF≜t^{T}\Phi_{s}^{-1}\left(x-m\right),\;GMF≜t^{T}\Phi_{G}^{-1}\left(x-m\right)\;,$$
where SMF and GMF refer to the segmented and the global version, respectively, $\Phi_{s}$ and $\Phi_{G}$ are the segmented and the global covariance, respectively, and the other parameters are as defined above.

This work focuses on the NSMF version, where each MF is normalized by its standard deviation. Additionally, its superior performance [8,10], it has the advantage of having its t component scaling invariant, as can be shown by rewriting the expression as follows:(3)$$NSMF≜\frac{t^{T}\Phi_{s}^{-1}\left(x-m\right)}{\sqrt{t^{T}\Phi_{s}^{-1}t}}={\tilde{t}}^{T}{\tilde{\Phi }}_{s}^{-1}\left(x-m\right);\;\tilde{t}=\frac{t}{\left|t\right|},\;{\tilde{\Phi }}_{s}=\Phi_{s}\cdot \sqrt{{\tilde{t}}^{T}\Phi_{s}^{-1}\tilde{t}}.$$

We can thus examine the target direction effects separately from its power. Therefore, in the sequel, the symbol *t* will assume a normalized vector.

### 1.3. Problem Statement and Prior Work

The effectiveness of segmentation in hyperspectral target detection remains a subject of ongoing debate, with outcomes often varying based on the target and data characteristics. For instance, in this study, segmentation applied to the same dataset and algorithm yielded divergent results: one target showed significant performance improvements with a higher NSMF ROC-Curve than the global one (NGMF), while another target showed no improvement despite identical conditions (Figure 1).

These observations suggest that segmentation efficacy is influenced not only by the data structure but also by other critical algorithmic factors. Indeed, the literature reflects this ambiguity: while segmentation sometimes improves detection [1,2,3], in other cases, it does not [4]. This paper builds on the foundational work of Ben Yakar [12] and provides an in-depth analysis of the simultaneous effects of key factors, including the target power (p), the covariance matrix (Φ), the target spectrum (t), and the error threshold (th). A comprehensive report, including mathematical analyses and experimental details, is available in the Appendix A.

Segmentation of hyperspectral datasets has been an active area of research for both classification and target detection processes [13,14,15,16,17], driven by the non-stationary nature of hyperspectral imagery in both spatial and spectral domains. Recent studies closely aligned with our focus have demonstrated the benefits of segmentation preprocessing for subpixel target detection. For example, Liang et al. [5,6] employed segmentation across spectral and spatial regions to detect both real and implanted targets, showing improved detection probabilities (see Figures 7 and 8 in [5]). Similarly, Stalley [7] compared various segmentation methods, including k-means, Gaussian mixture models, and subspace clustering, demonstrating that all methods outperformed non-segmented detection algorithms. These findings underscore the relevance and potential of segmentation preprocessing, further motivating its exploration in this study.

### 1.4. Evaluation Metrics

To quantify segmentation impact, we use two key metrics. The first metric is $A_{th}$, defined as:(4)$$A\left(th\right)≜\frac{\int_{0}^{th}P_{D}\left(\tau \right)d\tau -0.5\;{th}^{2}}{th-0.5\;{th}^{2}},$$
where $P_{D}$ represents the probability of detection, and th denotes the error threshold. Introduced by Caefer et al. [10], this metric measures the area under the ROC curve as a function of the maximum false alarm rate. It produces a scalar value between 0 and 1, with higher values indicating better average ROC performance.

The second metric is $B_{th}$, defined as:(5)$$B\left(th\right)≜\frac{A_{L}\left(th\right)}{A_{G}\left(th\right)},$$
where $A_{G}$ and $A_{L}$ refer to $A_{th}$ for a Matched Filter using a global or segmented estimated covariance matrix, respectively. Originally introduced by Ben-Yakar [12], this metric quantifies the improvement in detection performance attributable to segmentation. Higher values indicate greater benefits provided by segmentation.

### 1.5. Datacubes

This study employs a progressive reduction and deduction approach, starting from controlled synthetic scenarios and advancing towards fully real-world data. This methodology ensures robust and reliable conclusions applicable to practical hyperspectral target detection. This consists of three types of data, as depicted in Figure 2:(a)Synthetic dataset: This dataset consists of two standard Gaussian distributions with two covariance matrices (Φ_1_, Φ_2_) in two-dimensional space and a target *t* with power *p* Each Gauss represents the spectral cross-section of the second moment of the residual noise (*x* − *m*) of the corresponding segment and has 3 degrees of freedom: aspect ratio, scaling factor, and rotation angle. The target is represented as a 2D unit vector with two degrees of freedom—its magnitude and its rotation angle—corresponding to the target’s power and spectrum, accordingly. This minimalistic setup efficiently covers the range of possible target-to-background interactions while avoiding the complexities of high-dimensional data. This dataset serves as the fundamental building block for comprehensive synthetic simulations.(b)Simple In-house Cube (SIC): This 91-channel dataset contains stationary data with only two distinct areas. Its non-MVN distributions enable realistic simulations, bridging the gap between synthetic and real-world data. By manipulating the covariance matrices of two areas, we can simulate different types of inhomogeneity as interacted with any desirable target. This allows gradually increasing the complexity while retaining control over key parameters. The SIC serves as a baseline for exploring what can be practically deduced in more complex scenarios.(c)RIT Cube: This hyperspectral dataset depicts a scene around Cooke City, Montana, provided by the Rochester Institute of Technology (RIT) [18]. It contains highly non-stationary data with a mix of natural and manmade materials. The data were collected using a high-resolution imaging spectrometer under controlled conditions, with details on the camera, lens, and weather provided in Snyder et al. [18]. Real-world data from the RIT Cube closes the gap entirely by addressing the final layer of complexity, including multiple segments with natural spatial variability and diverse noise characteristics. These complexities allow for testing the full applicability of the guidelines and insights derived from the simpler datasets.

In addition, a second standard dataset, the Via Reggio dataset, provided by [19], was tested. This dataset features significantly different noise characteristics and environmental conditions and was used to validate the practical conclusions and guidelines presented in this paper. However, its results are omitted here due to redundancy.

A key aspect of the experimental framework in this study is the implantation of targets into datasets. This approach, described in Section 1.1, allows for systematic evaluation of algorithms under controlled conditions, enabling flexible testing of various target-to-background interactions. Unlike traditional methods relying on fixed, real-world targets, our strategy ensures that the interaction between targets and their backgrounds can be comprehensively analyzed. This methodology has been used and justified by other researchers [20].

For segmentation we utilized the K-Means algorithm [21], chosen for its simplicity and widespread use. This approach was sufficient for comprehensively studying the indirect effects of the various factors influencing segmentation efficacy. As a pixelwise classifier, K-Means does not account for spatial properties, focusing solely on spectral similarity. Considering the trade-off regarding the number of segments [22], we divided the SIC dataset into two segments and the RIT dataset into five segments, as shown in Figure 3.

## 2. Factors of Influence

This section analyzes how each of the involved factors affects the segmentation success in the order of the algorithm (Equation (2)): it starts with the effect of p, the target power, added to the pixel assuming an additive model. It continues with the effect of Φ, the covariance matrix directly affected by the data inhomogeneity. It continues with the effect of t, the target direction in the spectral domain. It then ends with the effect of th, the error threshold in terms of the number of false alarms.

### 2.1. Influence of the Target Power

This section analyzes the influence of the target power on segmentation’s efficacy.

#### 2.1.1. Effect on the Distributions

The expectation of the Matched Filter without a target present is zero. But when a target is added, a bias occurs:(6)$$\mu_{i}=p\cdot \sqrt{t^{T}\Phi_{i}^{-1}t},$$
where p and t refer to the target power and direction, respectively, and $\Phi_{i}$ refers to one of the global or local covariance matrices. Therefore, increasing the target power (p) increases the expectation proportionally, shifting the “with target” distributions to the right, providing more detections ($P_{D}$). Since this is true for any selected threshold, the performance (A) improves as well (see Equation (4)). Additionally, since Equation (6) holds for any kind of covariance, both the global and the local performance improve.

However, it is unclear what happens to the benefit (B), defined by the local to global performance ratio (Equation (5)), whether it remains constant or it grows in some way. Answering this requires delving into the role of p.

#### 2.1.2. Statistical Influence of the Target Power

Figure 2a may represent a spectral cross-section of two segments’ covariances, their composed covariance ($\Phi_{G}$), and a target vector (t) with some power (*p*). As such, the illustration shows that as *p* increases, the Signal-to-Noise Ratio (SNR) increases relative to any covariance, whether global or local. Equation (7) shows that this increase happens proportionally regardless of the domain:(7)$$SNR≜\frac{expected\;signal}{standard\;deviation}\underset{\underset{\left(2\right)}{↑}}{=}\frac{\mu }{\sigma }\underset{\underset{\left(3\right)}{↑}}{=}\mu \underset{\underset{\left(6\right)}{↑}}{\propto }p.$$

This insight leads to the intuition that there are two extremes and one intermediate state:When the target is weak, its SNR is bad even locally. There is, therefore, no point in segmenting since the resulting performance would remain bad in any case.When the target is strong, its SNR is already good globally, and segmentation is redundant.However, an intermediate *p* might exist, as in the example of Figure 2, where the SNR is still bad globally but already good locally. This case is where segmentation is worthwhile since it provides good local performance, which is, at the same time, significantly better than the global one.

Therefore, the response $B\left(p\right)$ is expected to be unimodal, low at the extrema, and maximal in the middle. Still, some characteristics of this shape are unclear, such as the location of the optimum, the maximal performance improvement, and the efficacy range of *p* values where segmentation is beneficial.

#### 2.1.3. Range of Effective Target Powers

A synthetic simulation, illustrated in Figure 4, enabled characterizing the shape of $B\left(p\right)$. Increasing p gradually from a tiny value showed that although the performance improves in both domains, a gap occurring between the global ($A_{G}$) and local ($A_{L}$) performance function causes the benefit function (B), given by their ratio (Equation (5)), to become bell-shaped. Below, the origin of this gap is analyzed through the performance functions.

It can be seen in Figure 4c that the local $A_{L}$ is the one that rises first, and therefore B initially rises (Figure 4d). This rising starts to become significant when the highest segment expectation in the local domain (Figure 4b) just crosses its decision threshold ($\eta_{L}$). This cross is related to the fastest increase in $P_{D}$, causing the local almost steepest ascent of $A_{L}$. Segmentation starts to be worthwhile.

As the target keeps getting stronger, B rises a bit more but reaches its maximum rapidly and begins to fall (Figure 4d). This fall happens due to its denominator $A_{G}$ that is starting to grow, thanks to the global distributions in which detection is beginning to be feasible (Figure 4a).

Thereafter, the benefit B keeps decreasing until becoming so low that segmentation is no longer worthwhile. This decrease comes from the steep ascent of $A_{G}$ (Figure 4c), originated in the global distribution (Figure 4a) that crosses its decision threshold ($\eta_{G}$).

This analysis reveals that the range of target powers in which segmentation might be effective is well defined by locating, both globally and locally, where the with-target expectation meets the decision threshold. We call these domain boundaries: $p_{1}$ after the leading local expectation and $p_{G}$ after the global expectation. Substituting Equation (6) gives explicit expressions for these indicators:(8)$$p_{1}≜\frac{\eta_{L}}{\underset{s}{\max}\sqrt{t^{T}\Phi_{s}^{-1}t}}\;,\;p_{G}≜\frac{\eta_{G}}{\sqrt{t^{T}\Phi_{G}^{-1}t}}.$$

These values could be viewed as anchors regarding p, since its impact on segmentation efficacy does not depend on its absolute value but only on its position relative to these values. As p is a multiplier, the degree of segmentation efficacy is indicated by the ratio of these anchors, that is,
(9)$$\Delta p≜\frac{p_{G}}{p_{1}}.$$

This index is called the “efficacy range’s width”.

#### 2.1.4. The Optimal Target Power

It can be seen in Figure 4d above that the benefit function B(p) becomes optimal shortly after $p_{1}$. Further experiments, summarized in Figure 5, show that this behavior is consistent for any type of data structure and that in the pure Gauss case, the optimal target power ($p_{max}$) satisfies approximately $p_{max}\approx \sqrt{2}\cdot p_{1}$.

Figure 5 also shows that the maximal benefit ($B_{max}$) increases together with the efficacy range (Δp). Their connection can be explained mathematically. At the $p_{max}$ point, $A_{L}$ is always 0.5, so the major effect on B comes from $A_{G}$ (Equation (5)). Since $\mu_{G}$ keeps decreasing, the numerator in Equation (4) tends to zero. Numerical approximations such as Taylor’s can show that this tendency decreases proportionally to $\mu_{G}$. Substituting Equation (8) and the $p_{max}$ approximation into Equation (6) gives $\mu_{G}\approx \sqrt{2}\cdot p_{1}\frac{\eta_{G}}{p_{G}}\propto \frac{1}{\Delta p}$. Compiling all together gives:(10)$$B_{max}\propto \frac{1}{A_{G}}\propto \frac{1}{\mu_{G}}\propto \Delta p.$$

The logic in this result is that the more the data structure challenges the detection due to its lack of homogeneity, the broader the range of targets to which segmentation can contribute. As such, relatively weaker targets become detectable, and segmentation brings a higher benefit.

#### 2.1.5. Principles

The target power (p) affects the segmentation benefit (B) through the SNR. It has a unimodal response that has a range of efficacy bounded by well-defined $p_{1}$ and $p_{G}$ anchors. The optimal target power goes with $p_{1}$ and the respective benefit is proportional to these anchors’ ratio (Δp).

### 2.2. Influence of the Data Inhomogeneity

This section analyzes the influence of the covariance matrix (Φ), mainly affected by the degree of inhomogeneity of the data from which it is estimated.

#### 2.2.1. A New Measure, “Kb”

The previous section concludes that the maximal segmentation benefit depends on the efficacy range (Δp), as defined in Equation (9). Substituting the anchors’ explicit expressions from Equation (8) gives a two-part expression, with the error threshold affecting only the first part, where the covariances and the target direction affect mainly the second part:(11)$$\Delta p=\frac{\eta_{G}}{\eta_{L}}\cdot \frac{\underset{s}{\max}\sqrt{t^{T}\Phi_{s}^{-1}t}}{\sqrt{t^{T}\Phi_{G}^{-1}t}}\Rightarrow K_{b}≜\frac{\underset{s}{\max}\sqrt{t^{T}\Phi_{s}^{-1}t}}{\sqrt{t^{T}\Phi_{G}^{-1}t}}.$$

The latter part is called $K_{b}$ after “benefit factor” since this scalar factor has a proportional effect on Δp, hence also on the maximal benefit.

Therefore, when the data structure changes and the covariances layout is affected, the segmentation performance is influenced by how this change is reflected through that key factor $K_{b}$. Therefore, it is essential to understand the significance of this new intermediate factor.

#### 2.2.2. Meaning of “Kb”

Substituting the Matched Filter expectation (Equation (6)) into $K_{b}$ definition (Equation (11)) gives a ratio between the maximal local expectation and the global expectation. However, $K_{b}$ might be better comprehended in the spectral domain where Equation (6)’s expression signifies “Mahalanobis Distance” along the target direction (t), as illustrated in Figure 6a:

Such a distance means “the length of the full target vector p⋅t, relative to its intersection point with the covariance matrix’s ellipsoid”, a point denoted here as XS. Hence there exists an inverse relationship between this intersection point (XS) and the expectation (μ) above. This connection derives that the $K_{b}$ expression can be rewritten as a ratio of intersection points:(12)$$K_{b}=\frac{\underset{s}{\max}\mu_{s}}{\mu_{G}}=\frac{{XS}_{G}}{\underset{s}{\min}{XS}_{s}}.$$

This expression gives, in a layout like the one illustrated in Figure 6b: the intersection with the global $\Phi_{G}$ relative to the local $\Phi_{1}$, or simply, $XS_{G}/XS_{1}$.

Two types of inhomogeneity are illustrated in Figure 7, with their corresponding $K_{b}$ annotated. Type (a) is “scaling”, where the segments’ covariances differ just by a scalar factor, and type (b) is “angular”, where the covariances differ by some planar rotation. In this illustration, despite the different inhomogeneity types, the $K_{b}$ ratio obtains a completely identical value. However, this identity sustains only in the specific illustrated angle of t, where in other angles, the $K_{b}$ ratio becomes different. This follows that $K_{b}$ reflects the degree of the data inhomogeneity, though not in the general sense, but rather in how it acts along the specific target direction.

#### 2.2.3. Principles

A newly introduced factor called $K_{b}$ was found to be a key in understanding the impact of variations in the data. That is since when the data structure varies, the covariances change by an influence that, to a specific target, is condensed into that scalar $K_{b}$, which has an immediate connection to Δp and affects the optimal benefit $B_{max}$ proportionally.

This new factor is given by a closed-form expression, much easier to calculate than the benefit that involves complex optimization over integrations. This expression is equivalently a ratio of expectations or a ratio of intersection points, indicating the degree of “directional inhomogeneity” of the data. Moreover, this expression implies that inhomogeneity does not determine the segmentation impact in general but rather its highest possible impact, where the specific impact depends on the specific target of interest.

### 2.3. Influence of the Target Direction

This section analyzes the influence of t, the target direction in the spectral domain.

#### 2.3.1. A Rule of Thumb: Theoretical Perspective

As discussed in Section 2.2.2, analyzing the target direction impact on the segmentation benefit is possible by tracking its effect on the intermediate factor $K_{b}$. Figure 8 below illustrates a case of angular inhomogeneity and a rotating target over some cross-section plane. The illustration shows that when t rotates, the ratio of the intersection points does not change significantly. Hence the maximal benefit $B_{max}$ is not changing significantly as a function of the target direction.

However, this is not the only possibility. Additional examples are shown in Figure 9. Specifically, similar to case (a), where $K_{b}$ is inherently rotation invariant, and case (b), which we have just analyzed, case (c) is also nearly invariant, as it represents a linear transformation of case (b); hence, proportions are preserved, and the $K_{b}$ ratio remains intact. However, cases (d) and (e) are different. In case (d), the covariances also have a scaling difference on top of just rotation. This scaling causes the target to achieve a $K_{b}$ ratio much smaller at 90° than at 0°. A similar phenomenon occurs in case (e), where only the major eigenvectors are rescaled.

This phenomenon reveals that it is likely to find the optimal target direction close to one of the strongest eigenvectors where the most significant gap between the global and the local intersections usually occurs.

#### 2.3.2. Estimating the Optimal Direction

The rule of thumb valid on two segments is not necessarily true on multiple segments. Fortunately, however, an analytical method for finding such a direction has been discovered. Formally, the optimal direction should be around the maximal $K_{b}\left(t\right)$, that is, in the direction where the highest ratio between $XS_{G}$ and the minor $XS_{s}$ occurs (Equation (12)). But since that minor value might jump from one local segment to another, it is easier to first focus on solving one single segment, $\Phi_{1}$, as in Figure 10a, in which the remaining objective is to maximize $XS_{G}/XS_{1}$.

It turns out that, in this reduced case, whitening the system by $\Phi_{1}$ provides an interesting result. As illustrated in Figure 10, the whitening transforms $XS_{1}$ to 1. In consequence, by Equation (12), $XS_{G}$ becomes none other than $K_{b}$ (Figure 10b). Since this property holds for any direction t, the whole ellipsoid of the whitened $\Phi_{G}$ represents $K_{b}$ in all possible directions. The biggest $K_{b}$ is therefore obtained where the largest radius occurs, which is simply on the major eigenvector (Figure 10c). All that remains, then, is to transform this vector back to the original domain, thus obtaining the optimal target direction in terms of $K_{b}$ (Figure 10d). In the general case, this process is repeated for every segment, and the highest $K_{b}$ is selected, as defined mathematically in the following equations:(13)$${\hat{t}}_{\max}≜\underset{t}{\mathrm{argmax}}\left\{K_{b}\left(t\right)\right\}=\Phi_{\hat{s}}^{0.5}\;{{\vec{K}}_{b}}_{max}^{\left(\hat{s}\right)},\;\hat{s}=\underset{s}{\mathrm{argmax}}\left\{\left\|{\vec{K}}_{max}^{\left(s\right)}\right\|\right\},\;{{\vec{K}}_{b}}_{max}^{\left(s\right)}≜\underset{t}{\max}{\vec{K}}_{b}^{\left(s\right)}=\pm \mathrm{major}\left(\Phi_{s}^{-0.5}\Phi_{G}\Phi_{s}^{-0.5}\right)\;.$$

It is worth noting that the proposed process relies solely on covariance matrices and involves highly efficient operations, making it computationally fast. Any optimal direction ${\hat{t}}_{\max}$ is only defined up to a sign (±t).

On real data, maximizing $K_{b}$ provides only an estimation of the optimal direction. Where this is generally reasonable in predicting the maximal efficacy width (Δ*p*), due to their direct relationship (Equation (11)), it is less accurate in predicting the maximal benefit. Not only that, the actual benefit at $\pm {\hat{t}}_{\max}$ may vary significantly due to asymmetries in the data structure, but the real maximal benefit ($B_{max}$) usually does not occur exactly there, as this estimation is based on the second moment and does not fully capture the real data distribution. Similarly, the range of effective target directions can be approximated by solving $\underset{t}{\arg}K_{b}\left(t\right)>\beta$ for a desirable threshold β.

#### 2.3.3. Principles

The target direction t interacts with the covariances, affecting segmentation efficacy through the key factor $K_{b}$. The optimal direction can be estimated analytically by maximizing $K_{b}$, which gives a closed-form solution, easy-to-calculate. Likewise, the directions’ efficacy range can be estimated by comparing $K_{b}$ to some desirable threshold.

### 2.4. Influence of the Decision Threshold

This section analyzes the influence of th, the error threshold, in terms of the number of false alarms.

#### 2.4.1. The Joint Impact of the Error Threshold

According to Equation (1), permitting fewer false alarms by a lower error threshold (th) derives a higher decision threshold (η). Since this property holds for both the global and the segmented MF, both $p_{1}$ and $p_{G}$ anchors increase proportionally (Equation (8)). Hence, while the optimal power ($p_{max}$), which goes with the first anchor, should increase (Section 2.1.4), the efficacy width (Δp), given by the ratio of these anchors (Equation (9)), is not expected to change significantly. However, the target direction (t) operates orthogonally, since its impact on the performance passes mainly through the $K_{b}$ factor, which is independent of the thresholds (Equation (11)).

A synthetic simulation, shown in Figure 11, demonstrates the effect of reducing th threshold. Indeed, the optimal target power ($p_{max}$) shifts right along with both anchors, while the relative width (Δp) remains constant. Surprisingly, yet, along with them, the optimal benefit ($B_{max}$) increases, instead of remaining constant after its proportional relation to Δp (Equation (10)).

#### 2.4.2. Impact on the Maximal Benefit

Understanding the root of $B_{max}$’s rising was possible using a synthetic simulation, illustrated in Figure 12. In that simulation, th was reduced gradually from a high value, causing η to shift right in both the global and the local distributions. Simultaneously, for tracking the impact on $B_{max}$, the target power was adjusted as to sit on its optimal point ($p_{max}$). Since such an adjustment causes the highest local expectation to sit consistently just after the local threshold (Figure 12b), $P_{D}$ barely changes, and the local performance ($A_{L}$) remains almost constant (Figure 12c). Contrarily, globally, while the threshold shifts similarly, the with-target distribution lags behind (Figure 12a), $P_{D}$ decreases consistently, the global performance ($A_{G}$) drops monotonically (Figure 12c), and the benefit function grows endlessly (Figure 12d). Thus, unlike B(p), which responds unimodally (Figure 4), th has a monotonic response.

Intuitively, this behavior indicates that as the global detection becomes more challenging due to a stricter threshold, the impact of segmentation becomes more pronounced. Nevertheless, it remains intriguing why the behavior differs so significantly between the global and local domains. After all, shifting a threshold is equivalent to inversely shifting expectations, so one would expect the change in the number of detections to change similarly in both domains.

Delving into the above theory reveals that due to Equation (1), reducing the error threshold shifts both the global and the local decision thresholds in similar proportions. Whereas, as a target power increases, each global or local expectation grows relative to the intersection point in its domain (Figure 6), whose ratio is nothing but $K_{b}$ (Equation (12)). The simulation above (Figure 12) used $K_{b}=8$, which caused the global expectation to barely move, although the respective threshold kept shifting. Thus, unlike expectations, changing a target power is inequivalent to an opposite shift of the respective decision threshold. This insight means that the global domain, where the target is weaker relative to the reference noise, requires more relative efforts to improve the SNR.

#### 2.4.3. Impact per Domain

The definite trend of the optimal point can be extracted from mathematics. The optimal power ($p_{max}$) goes with the first anchor (Section 2.1.4), which substituting Equation (9)’s components with Equation (11) and Equation (8) gives a proportion of $\eta_{L}/K_{b}$ for any given global SNR. The corresponding benefit ($B_{max}$) evolves with $A_{G}$, due to the constant $A_{L}$ (Figure 12c), hence it depends on an integration over $P_{D}$ in the global domain (Equation (4)). Since $P_{D}$ is based on the with-target distribution ($f_{T}$), which is a shifted version of the without-target distribution ($f_{N}$), its integration gives a similar response as $P_{FA}$ which gives 0.5$th^{2}$ [10]. This implies that $A_{G}$ in Equation (4) is proportional to $th^{2}/\left(th–0.5th^{2}\right)$ which tends to th as th goes to zero. Thus, $B_{max}$ tends to the inverse of th. Additionally, Equation (11) gives $\Delta p\propto K_{b}$ which, substituting into (10) gives $B_{max}\propto K_{b}$. Combining all together gives, asymptotically, the following proportions:(14)$$p_{max}\;,p_{1}\propto \frac{\eta_{L}}{K_{b}},\;B_{max}\propto \frac{K_{b}}{th}.$$

Interestingly, the threshold effect is related to the domain of each factor: the target power factors vary with η that is in the scores’ domain, where the optimal benefit varies with th which acts in the integral domain.

#### 2.4.4. Principles

The more challenging the detection becomes due to a tighter threshold, the more the $p_{1}$ and $p_{G}$ anchor points increase, as well as the position and intensity of the optimal benefit. The response to such a change is monotonic and tends to evolve proportionally, unlike the target power that responds unimodally. The threshold has a negligible effect on the target’s efficacy range and has an orthogonal effect to $K_{b}$ and thus to any of the target direction factors.

### 2.5. Guidelines

This section analyzes how each of the NSMF factors affects the segmentation success:The target power (p) affects the efficacy of segmentation through the SNR. The segmentation’s benefit varies unimodally with respect to two key anchors called $p_{1}$ and $p_{G}$. The optimal p is proportional to the small anchor ($p_{1}$), while the width of the efficacy range is determined by the ratio of $p_{G}$ to $p_{1}$ (Equation (9)).The covariances ($\Phi_{s}$), reflecting variations in the data structure, affect segmentation’s efficacy through a scalar key factor named $K_{b}$. This factor qualifies “directional inhomogeneity” along the target direction and has a proportional influence on both the targets’ efficacy range (Δt) and the maximal benefit ($B_{max}$).The target direction (t) interacts with the covariances, affecting efficacy through $K_{b}$ factor. Maximizing $K_{b}$ enables estimating the optimal direction using a closed-form expression that is computationally straightforward.Threshold (th): Tightening the threshold raises both $p_{1}$ and $p_{G}$ anchors, resulting in a monotonic increase in the optimal segmentation benefit. Such a change triggers proportional relationships: the target power factors vary with the local decision threshold ($\eta_{L}$), while the optimal benefit varies with the error threshold (th).

These properties provide a framework for characterizing practical cases, allowing for informed decision-making regarding the range of targets and thresholds where segmentation is worthwhile in the detection process.

## 3. Experiments

This section presents the experimental results conducted to validate the theoretical conclusions. It begins by introducing the preparations made for these experiments. Subsequently, it delves into the influences of the data structure on the results. Finally, it explores the influence of the target signature on real datasets.

### 3.1. Experiments Considerations

For our experiments, we utilized the three types of data described in detail in Section 1.5: the Synthetic Dataset, the Simple In-House Cube (SIC), and real-world data, represented by the RIT Cube. Each dataset offered unique characteristics and challenges for experimentation. To ensure the reproducibility of our results, this section outlines the special preprocessing steps applied to prepare these datasets for experimentation.

Each of these three datasets posed unique preprocessing challenges: The Synthetic Dataset required adjustments to maintain analytical consistency. The SIC dataset involved targeted manipulations to introduce controlled inhomogeneities. Finally, the RIT Cube required addressing significant boundary estimation biases caused by the non-convexity of real-world segments.

The principal methods used to address these challenges are outlined below. A more detailed description is available in the comprehensive study report referenced in the Appendix A.

#### 3.1.1. Preprocessing

Several preprocessing steps were necessary for SIC data simulations. One critical step involved ensuring that data manipulations were based on a neutral starting point. First, we addressed the imbalance in segment sizes by cropping the image from 50 × 50 to 34 × 47, effectively reducing the populations imbalance from 40% to 0.2%. Next, we addressed stationarity by balancing the segment covariances using a unique whitening transform, defined as $\Phi_{G}^{0.5}\Phi_{s}^{-0.5}$. This process whitens segment s by its own covariance and then unwhitens it by the global covariance. As a result, this manipulation effectively reduced the covariances’ deviation from 5° to 0°.

The SIC data simulations were carried out over cross-sections of interest, for example, by the plane spanned by the major and minor $\Phi_{G}$’s eigenvectors. Such a simulation includes linear transformations for each segment across planar sections by transforming the data by $VTV^{T}$, where V is a basis, such as the eigenvectors matrix, and T is a desirable transform matrix. The latter is the identity matrix, excluding four cells determining a 2D transform over the two selected dimensions.

#### 3.1.2. Practical Aspects

Each type of data had its challenges after its specific characteristics and its respective processing.

In some extreme cases, the synthetic simulations encountered numerical errors. To address this, a more accurate method was developed, incorporating three key principles: 1. maintaining pure analytic forms consistently, 2. employing standard numeric solutions for Gaussian distributions, and 3. applying approximation such as L’Hôpital’s theorem.

In the SIC realistic simulations, the presence of long-tailed distributions impacting performance, coupled with data quantization leading to high volatility, posed challenges in deriving reliable conclusions. To address this issue, we compared each result with a corresponding synthetic simulation, enabling us to characterize distribution shape properties and meticulously isolate stochastic phenomena.

In RIT’s standard dataset, background estimation errors occurring along the boundaries of segments introduced a notable bias in the per-segment average noise (x − m). This challenge was successfully resolved by developing a unique mirror padding technique, specifically designed for non-convex borders, as elaborated in Appendix B.

#### 3.1.3. Consistency

Special attention was paid to isolating the primal sources of influence. While target power exclusively affects the SNR (Equation (7)), data structure and target direction simultaneously affect both homogeneity (Equation (11)) and SNR (Equation (6)). To understand their roles thoroughly, we fixed the SNR while varying the data structure and the target direction. This was obtained by consistently normalizing the vector t so that the global expectation ($\mu_{G}$) remains constant.

### 3.2. Influence of the Data Structure

This section presents results regarding the influence of inhomogeneity due to variations in the data structure. It presents representative results for each of the three types of datacubes.

#### 3.2.1. Synthetic Simulation

Using the synthetic dataset, we gradually rescaled two identical segments. In consequence, the inhomogeneity increased and $K_{b}$ grew, respectively. Figure 13a shows the effect of $K_{b}$ on the optimal target power ($p_{max}$), in the upper graph, and on the respective benefit ($B_{max}$), in the lower graph, both on the logarithmic scale. The results show that for a high enough $K_{b}$, $p_{max}$ decreases proportionally and $B_{max}$ increases proportionally. Additionally, as the error threshold (th) decreases, both $p_{max}$ and $B_{max}$ grow consistently.

Although these results match the theoretical conclusions of Equation (14) asymptotically, it seems that at $K_{b}<10$ the benefit grows faster. For example, at a threshold of 0.01, there is a quadratic growth rate of $K_{b}^{2}$. An alternative model that approximately satisfies the plots might be $B_{max}\;\approx {\;H}_{10}\left(K_{b}\right)\cdot K_{b}/10th+{\bar{H}}_{10}\left(K_{b}\right)\cdot th^{-\log K_{b}}$, where $H_{10}(\cdot )$ and ${\bar{H}}_{10}(\cdot )$ refer to a composition around 10, such as the Heaviside unit function $u(K_{b}-10)$ and its complementary, respectively. However, such a model would only be applicable for data that follows normal distributions, which is not typically the case [11].

#### 3.2.2. SIC Real Simulation

Using the SIC dataset, we repeated the simulation for real data. Starting with a purely stationary state, we gradually rescaled the two segments by some multiplicative factor so that $K_{b}$ increases, regardless of the target direction. Figure 13b shows the effect on the optimal target power $p_{max}$ and on the respective benefit.

The results closely resemble those in the synthetic case above, with the primary distinction being that $B_{max}$ converges to a constant value. The origins of this difference can be traced back to quantization and the presence of long-tailed distributions in real data. In extreme $K_{b}$, the optimum ($p_{max}$) is obtained at a super weak target that shifts the global expectation to almost zero. Then, due to quantization, the number of detections becomes like the number of false alarms, the global performance stops falling, and the benefit ($B_{max}$) stabilizes at a fixed value.

#### 3.2.3. RIT Standard Dataset

In the RIT standard dataset, the underlying structure is inherent. However, since $K_{b}$ is related to the interaction with the target direction (Equation (11)), the data can be examined by selecting significant directions. We, therefore, chose the directions of the strongest eigenvectors from each of the five covariances of this data. For each direction, we computed $K_{b}$, determined the optimal p along with its respective benefit ($B_{max}$), and plotted it on scattergrams as shown in Figure 13c.

The results exhibit behavior consistent with the previous findings: as $K_{b}$ rises, $p_{\max}$ drops, and $B_{max}$ grows proportionally. The progression of $B_{max}$ follows an approximately quadratic rate in this specific case, as $K_{b}$ is less than 10, and as we chose a threshold (th) of 0.01.

### 3.3. Influence of the Target Signature

This section presents results regarding the influence of the target signature on real datasets. It starts with a representative two-segment case. It then continues with a standard dataset and examines diverse types of target spectrum, striving to deduce ways to improve the performance further.

#### 3.3.1. A Rule of Thumb: Practical Perspective

To test the target direction impact in the reduced two-segment case discussed above (Section 2.3.1), we used the SIC real dataset and a cross-section plane spanned by the major-to-minor $\Phi_{G}$’s eigenvectors (Section 3.1.1). Over that plane, we simulated angular inhomogeneity and rotated a target from 0° to 90° (Figure 14a). We then measured the optimal benefit obtained at each angle. The resulting upper graph in Figure 14c shows almost no rotation impact.

Thereafter, we gradually reduced the major eigenvector of $\Phi_{2}$ and rechecked the rotation impact (Figure 14b). The two lower graphs in Figure 14c exhibit a decline as they approach 90°, in accordance with the decreasing factor. This decline happens due to the reduction in the ratio of $K_{b}$ intersections as explained in Section 2.3.1. This reveals that since such asymmetries are common, it is likely to find the optimal target direction close to one of the most dominant eigenvectors.

#### 3.3.2. Estimating the Global Optimum

To examine the impact of different target directions in the standard RIT dataset, we started with searching for the global optimal target ($t_{max}$). We first estimated this optimum by maximizing $K_{b}$ using Equation (13). To get a sense of how good this estimate is, we crossed the spectral space with a plane spanned by the estimated ${\hat{t}}_{max}$ vector and $\Phi_{5}$’s major eigenvector (Figure 15a). Over this plane, we rotated a target and measured $K_{b}$ and $B_{max}$ per angle.

The results appear in Figure 15b. The lower graph shows $K_{b}$, which is indeed maximal at 0°, where ${\hat{t}}_{max}$ sits. The upper graph shows the actual $B_{max}$, whose true maximum is not much higher than its value at ${\hat{t}}_{max}$, but still, its position is somewhat shifted to the left. Fortunately, however, the curve between these two points looks convex, implying that the global maximum might be achieved using standard optimization methods initialized at the estimated point.

Table 1 column (a) shows the performance values for the estimated best direction. For a 0.01 error threshold, segmentation improves performance from 0.006 to 0.252, which is 40.6 times better. By comparison, the best target found in Ben Yakar’s work improved the performance by only 4.1 times (see Table 1 in [12]). Table 1 also shows that a smaller error threshold improves even more dramatically. So, our method seems to enable finding a direction with a benefit from the highest possible, which is also close enough to the absolute maximum.

#### 3.3.3. Limiting the Optimum to the Positive Cone

Unfortunately, the just found global ${\hat{t}}_{max}$ cannot be a target spectrum since some of its components are negative. In Figure 16a, the area outside the positive cone is grayed out, showing that ${\hat{t}}_{max}$ remains outside, while $\Phi_{4}$’s eigenvector remains inside the cone. Figure 16b shows that the optimal direction along the rotation towards that eigenvector sits precisely on the cone border, making it relatively easy to find.

Table 1 column (b) shows that even under the positivity constraint, segmentation improves performance by 36.4 times, which is still remarkable. So, rotating the global best direction towards a target that sits within the valid area enables finding a direction that meets the required constraint but is still close enough to the absolute maximum.

#### 3.3.4. Limiting the Optimum to the Image Pixels

Since forcing positivity is not enough to promise a valid signature, we additionally narrowed the constraint to the set of the image pixels so that the target would be undoubtedly valid. We searched for the pixel whose direction provides the maximal benefit among all the pixels. Figure 17a shows a map of all the pixels that give $K_{b\;}>\;2$. We analyzed each of these pixels using our rules and found the pixel that offers the maximal benefit. This pixel is annotated “×” in three different views in Figure 17. Notably, this pixel belongs to segment #5, which boasts the highest benefit among the five segments, where the pixel itself is positioned on the border of that segment and visually differs from most other pixels within this segment.

Table 1 column (c) shows that despite the extreme constraint, segmentation enhances performance by 13.2 times, which is still three times better than Ben-Yakar’s best result [12]. Hence, one might get a sense of the maximal benefit of valid targets through image self-analysis of the kind presented here.

#### 3.3.5. Efficacy with the Provided Targets

We further constrained the selected target by narrowing the possible set to only the twelve targets provided as part of the dataset. The best performer was a laboratory sampled blue cotton (F3). The result, shown in Table 1 column (d), gives that for a 0.01 error threshold, segmentation improves by only three times. This result indicates that segmentation might be redundant for such targets.

Overall, the results in Table 1 show that although segmentation provides a high impact in some directions—while narrowing the set of valid targets, this impact drops until becoming negligible. This rapid degradation implies that the efficacy range of valid directions might be relatively tight.

#### 3.3.6. Directions’ Efficacy Range

One option for getting an impression of the efficacy range is sampling targets from the dataset itself. Figure 17a depicts such an example, showing a map of $K_{b}>2$ pixels. It was found to be 97.5% accurate (measured by the relative number of true hits) equivalent to Δp>3.98 and 95.6% accurate equivalent to $B_{max}>3.98$, except that its computation lasts only 1.5 s instead of more than 100 h. Curiously, in this resulted map, the lit pixels occupy only 7.4% of the image. This result means that not many valid directions exist where segmentation is beneficial for this specific dataset.

One of the reasons for the limited domain of targets is the high sensitivity to the exact direction. It can be shown by comparing $\Phi_{5}$’s major eigenvector to the pixel with the most correlated signature, which, despite the considerable similarity (Figure 18a), tracking their cross-section reveals that $B_{max}$ drops from 17 to 4.7 in just 2.3° (Figure 18b). This phenomenon might be explained by the above insight that $K_{b}$, from which $B_{max}$ evolves, is represented by the radius of a multidimensional ellipsoid in the whitened domain (${\tilde{\Phi }}_{G}$ in Figure 10c). It turns out that, like in the data [1], the first eigenvectors of this ellipsoid are relatively strong and then weaken rapidly. Therefore, when the target rotates between strong eigenvectors, $K_{b}$ barely changes. Whereas, when it rotates towards most of the other weak dimensions, then a rapid degradation occurs. This behavior implies that the efficacy range of target directions is built from a manifold wide in few dimensions but narrow in most others.

### 3.4. Efficacy with RIT Data

According to the results, discerning the worthiness of segmentation for the RIT dataset poses an initial challenge. On one hand, the observed benefits are pronounced in a limited range of targets; however, on the other hand, this range broadens as the required threshold diminishes, amplifying the overall segmentation’s benefit. Yet, a pivotal consideration in this example may stem from the utilization of a basic “K-Means” segmentation approach, which might not inherently prioritize the creation of more homogeneous areas, contrary to our initial objective. Therefore, a transition to a more suitable technique designed to enhance stationarity might potentially result in a substantial enhancement to the detection across a broader range of targets, even under stringent threshold conditions.

## 4. Conclusions

This study aimed to identify the conditions under which segmentation improves the detection of subpixel targets in hyperspectral images. While data inhomogeneity was initially hypothesized as the primary factor, our comprehensive analysis revealed that multiple algorithmic factors also significantly impact detection performance.

Our analysis of the NSMF algorithm identified the matched target as a critical factor in segmentation performance, specifically through its influence on “directional inhomogeneity”. We introduced a novel factor, “$K_{b}$”, representing the maximal benefit, and developed a closed-form estimator for determining the optimal target direction. These findings were distilled into practical rules describing the influence of each factor on segmentation performance.

We validated these insights through extensive simulations across a broad range of conditions. Additionally, we applied these findings to two standard datasets, evaluating segmentation performance for various targets and characterizing the conditions under which segmentation is beneficial. This enabled deducing properties about the setup and proposing ways to enhance the detection. Further investigations into additional datasets and scenarios are ongoing.

In contrast to prior studies that primarily demonstrated segmentation’s potential for improving detection performance [5,6,7], this work focused on dissecting the underlying factors influencing segmentation efficacy. By analyzing these factors systematically, this study bridges the gap between observed benefits and the conditions that enable them.

Additionally, the theoretical insights, this work introduces practical analysis tools that can support future research and real-world applications, including:p-rules: Essential for characterizing segmentation’s effects on detection performance.$K_{b}$: A key factor for efficiently predicting segmentation benefit for any target.${\hat{t}}_{max}$: A closed-form estimator for the optimal target direction, easily computable.

These tools can be adapted for use with different models and algorithms. Unlike our prior work [8], which focused only on the influence of the matched target, this study provides an in-depth analysis of all contributing factors, supported by theoretical proofs and comprehensive experimental validations.

This study lays the groundwork for completing prior research on predicting segmentation benefits a priori. While an earlier study proposed linking data inhomogeneity to segmentation efficacy, we lacked a practical framework due to the complexity of segmentation-dependent factors. The $K_{b_{max}}$ metric introduced here simplifies this challenge by providing a single, interpretable measure of inhomogeneity for any given segmentation. This enables systematic comparisons with segmentation-independent inhomogeneity metrics, allowing evaluation of their ability to predict segmentation’s potential benefit.

This study focused on the Normalized Segmented Matched Filter (NSMF) algorithm to analyze segmentation efficacy. Ongoing research is testing whether segmentation is similarly effective in other algorithms, such as the Adaptive Cosine Estimator (ACE), which also relies on covariance-based whitening. Additionally, future work will explore alternative segmentation strategies, including hierarchical clustering, to assess their impact on the conclusions drawn here. These efforts aim to validate and expand the applicability of the proposed framework across different detection algorithms and segmentation methods.

## Figures and Tables

**Figure 1 sensors-25-00272-f001:**
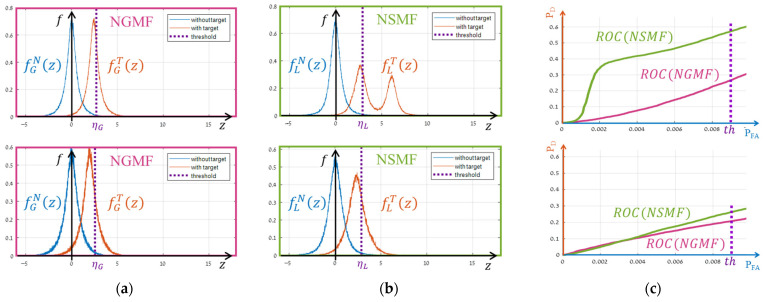
NSMF results on real data for two targets: the top row shows a real case of improved detection with segmentation, while the bottom row shows no improvement despite using the same data and segmentation. Column (**a**) displays global distributions, column (**b**) shows local distributions, and column (**c**) presents the ROC curves. Subscripts G /L denote global/local distributions, and superscripts N/T represent distributions without/with a target, respectively.

**Figure 2 sensors-25-00272-f002:**
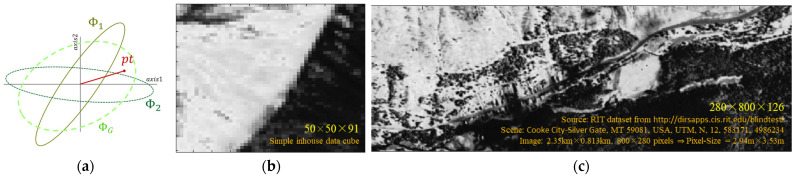
The three datasets used for this study: (**a**) synthetic data of Gaussian distributions, (**b**) one band from a Simple In-house Cube (SIC), and (**c**) one band from a data-cube, provided by Rochester Institute of Technology (RIT).

**Figure 3 sensors-25-00272-f003:**
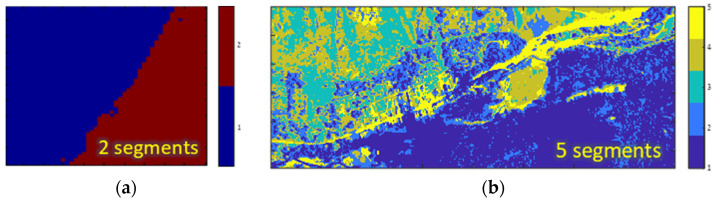
Segmentation maps used for (**a**) SIC datacube and (**b**) RIT datacube. The labels’ indexes correspond to the local covariances used in the sequel, that is, $\Phi_{1}$ corresponds to segment #1, $\Phi_{2}$ to segment #2, etc.

**Figure 4 sensors-25-00272-f004:**
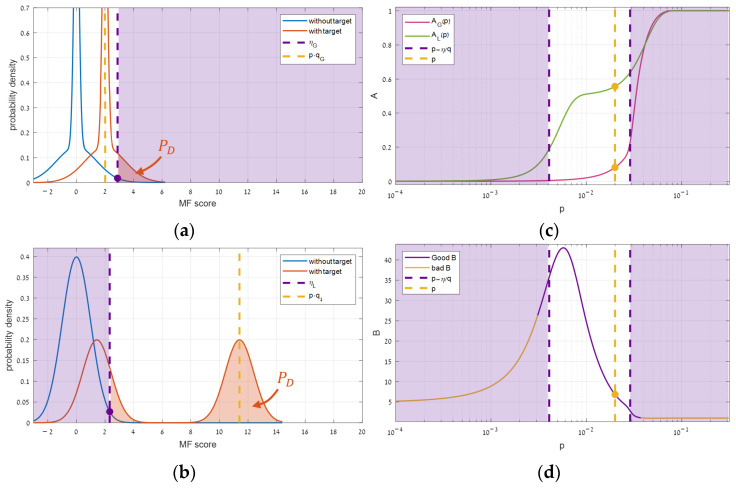
Range of effective target powers through different domains: (**a**) the global distributions, (**b**) the local distributions, (**c**) the respective performance, and (**d**) the resulted segmentation benefit. The purple grayed-out region represents the powers that are out of the efficacy range.

**Figure 5 sensors-25-00272-f005:**
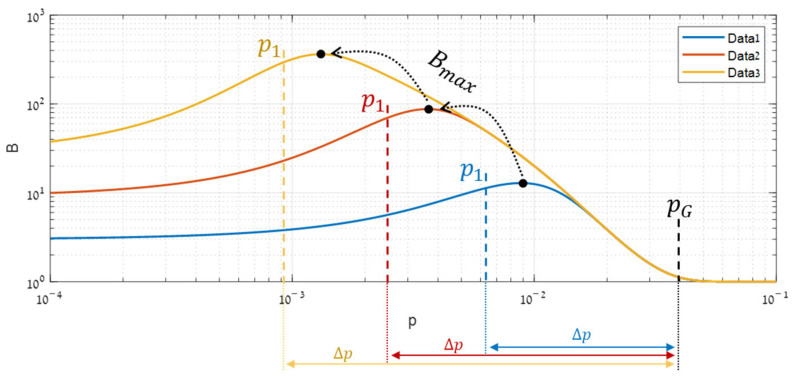
Evolution of B(p) as a function of inhomogeneity. The global SNR is fixed, hence, $p_{G}$ remains constant.

**Figure 6 sensors-25-00272-f006:**
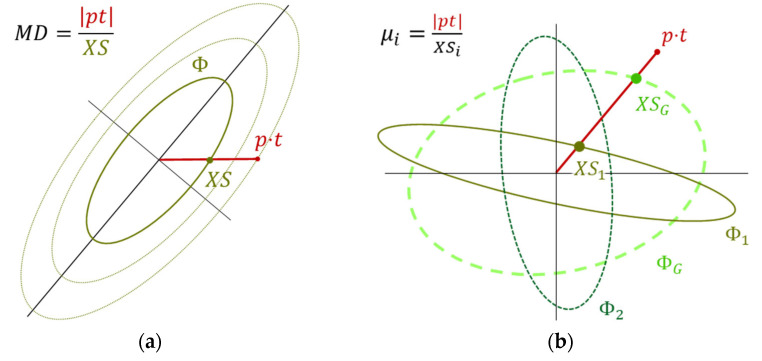
The spectral meaning of $K_{b}$: (**a**) expectation as a Mahalanobis Distance and (**b**) ratio of intersection points.

**Figure 7 sensors-25-00272-f007:**
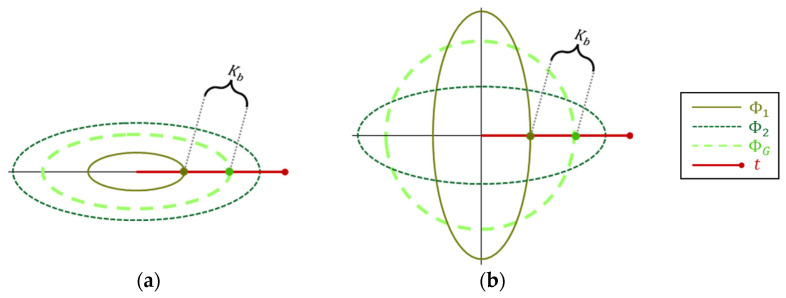
Comparing $K_{b}$ on two kinds of inhomogeneities: (**a**) scaling and (**b**) angular.

**Figure 8 sensors-25-00272-f008:**
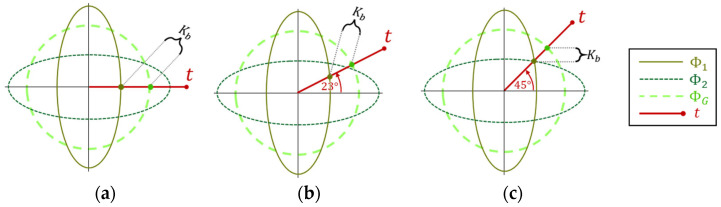
Angular inhomogeneity through a cross-section plane and a target at three angles: (**a**) 0°, (**b**) 23°, and (**c**) 45°.

**Figure 9 sensors-25-00272-f009:**
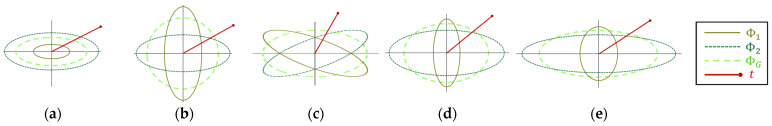
Target direction impact on different layouts: (**a**) scaling, (**b**) $\Phi_{1}$ rotation; upon it, (**c**) appends system rotation and scaling, (**d**) appends rescaling of both ellipsoids, and (**e**) appends rescaling of the major axis of both ellipsoids.

**Figure 10 sensors-25-00272-f010:**
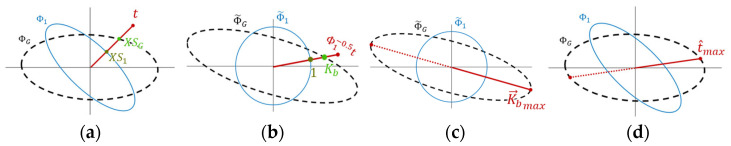
Analytic process of maximizing $K_{b}\left(t\right)$: (**a**) the objective with respect to one segment, (**b**) the effect of whitening, (**c**) the optimal $K_{b}$ in the whitened domain, and (**d**) the equivalent in the original domain.

**Figure 11 sensors-25-00272-f011:**
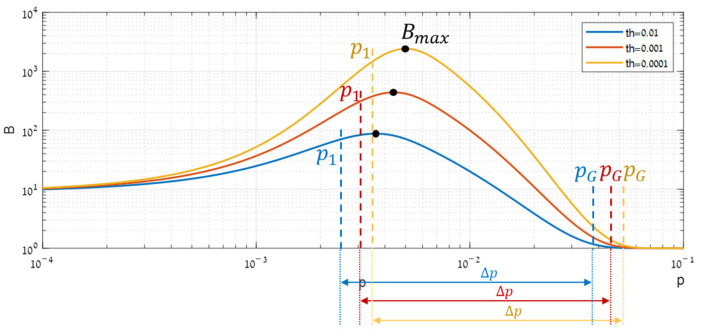
Evolution of B(p) as a function of different thresholds.

**Figure 12 sensors-25-00272-f012:**
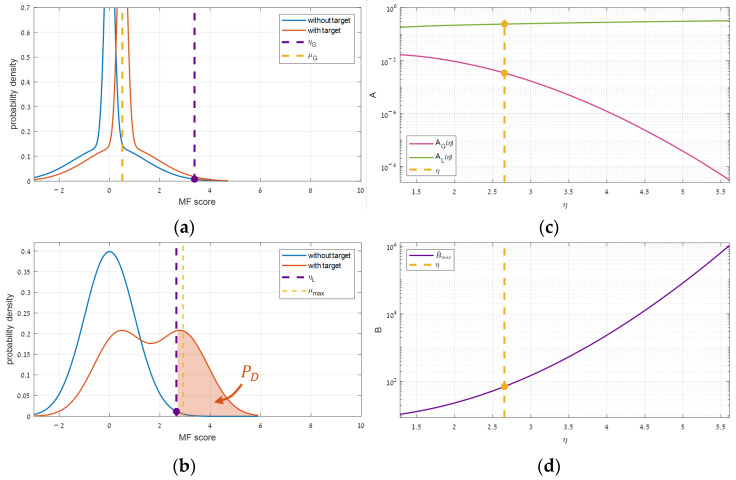
The influence of the error threshold in different domains: (**a**) the global distributions, (**b**) the local distributions, (**c**) the respective performance, and (**d**) the resulted segmentation benefit.

**Figure 13 sensors-25-00272-f013:**
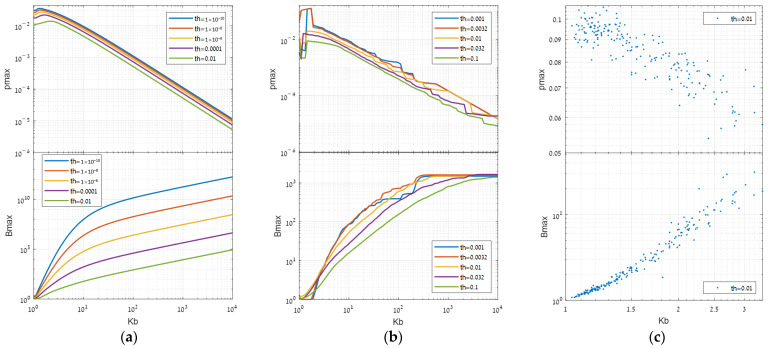
Influence of inhomogeneity for different thresholds on different types of datasets: (**a**) synthetic, (**b**) SIC cube, (**c**) RIT cube.

**Figure 14 sensors-25-00272-f014:**
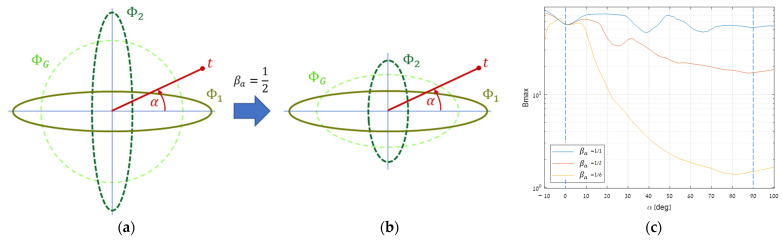
Influence of the target direction in the two-segment case on a representative layout: (**a**) the layout, (**b**) after reducing one major eigenvector by half, and (**c**) the resulting maximal benefit in three cases.

**Figure 15 sensors-25-00272-f015:**
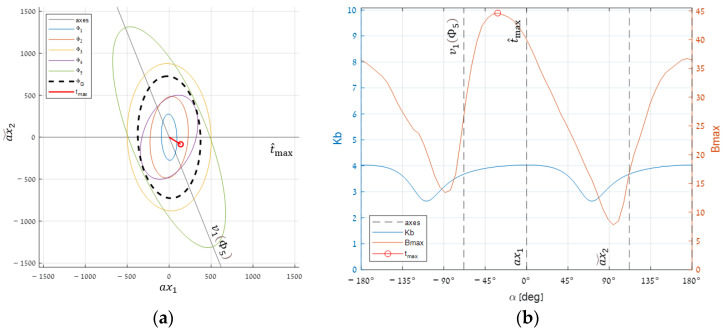
Segmentation performance for linear combinations of ${\hat{t}}_{max}$ with $\Phi_{5}$’s major eigenvector ($\nu_{1}$): (**a**) a planar cross-section for the selected pair and (**b**) segmentation performance for a range of angles.

**Figure 16 sensors-25-00272-f016:**
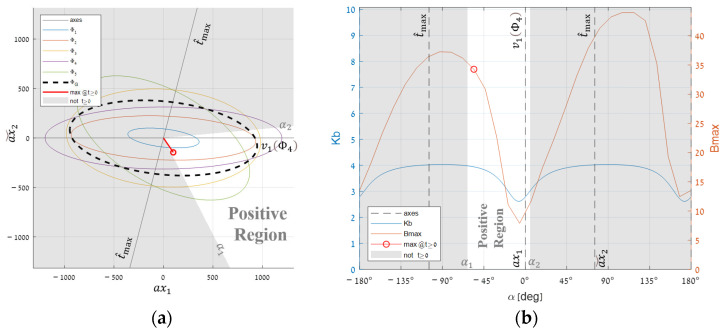
Segmentation optimum for constraint combinations of ${\hat{t}}_{max}$ with $\Phi_{4}$’s major eigenvector ($\nu_{1}$): (**a**) a planar cross-section for the selected pair and (**b**) segmentation performance for different angles.

**Figure 17 sensors-25-00272-f017:**
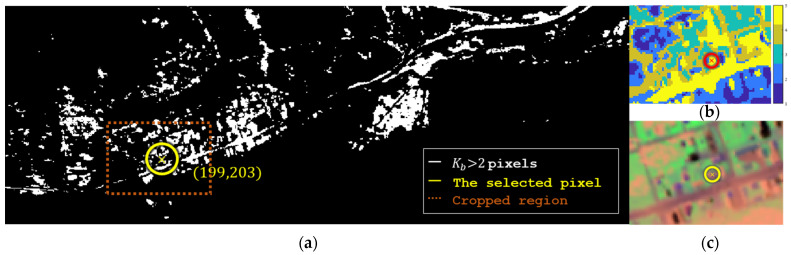
Several views with the best performing pixel, annotated “⊗”: (**a**) a map of all $K_{b\;}>\;2$ pixels, (**b**) the segmentation map cropped at (**a**)’s dashed region, and (**c**) a PCA-based visualization of the cropped datacube.

**Figure 18 sensors-25-00272-f018:**
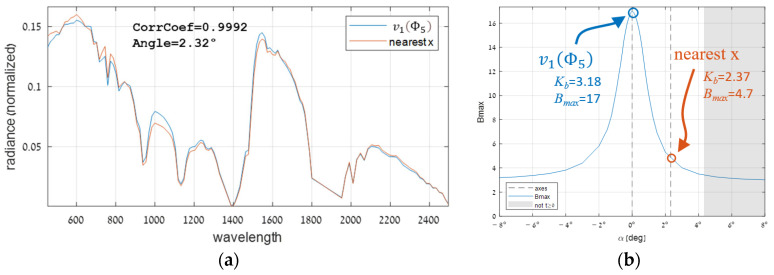
The performance of $\Phi_{5}$’s major eigenvector compared to its most similar pixel: (**a**) comparison of their two signatures and (**b**) performance analysis.

**Table 1 sensors-25-00272-t001:** RIT cube performance obtained with the optimal target under different constraints: (a) no constraint, (b) within the positive cone, (c) belongs to the data, and (d) belongs to a given set of real targets.

Constraint	$(a)\;t_{max}$			(b) t≥ 0			(c) t∈data			(d) t∈targets		
th	**0.001**	**0.01**	**0.1**	**0.001**	**0.01**	**0.1**	**0.001**	**0.01**	**0.1**	**0.001**	**0.01**	**0.1**
$K_{b}$	4.03			3.92			2.91			1.63		
Δp	4.52	4.06	3.58	4.18	3.91	3	3.32	2.52	2.44	1.7	1.6	1.53
$p_{\max}$	0.104	0.05	0.018	0.17	0.06	0.016	0.22	0.05	0.02	34 × 10^−4^	13 × 10^−4^	6 × 10^−4^
**NGMF (** $A_{G}$ **)**	6 × 10^−4^	0.006	0.039	6 × 10^−4^	0.006	0.033	7 × 10^−4^	0.012	0.084	0.005	0.037	0.149
**NSMF (** $A_{L}$ **)**	0.273	0.252	0.241	0.257	0.218	0.201	0.224	0.154	0.202	0.135	0.123	0.208
**Benefit (** B **)**	453	40.6	6.11	451	36.4	6.1	334	13.2	2.42	26.8	3.32	1.4

## Data Availability

The synthetic data used in this study are generated programmatically using deterministic parameters, and the code for their generation is provided in the repository associated with this article. The RIT dataset is publicly available at http://dirsapps.cis.rit.edu/blindtest/, while the Via Reggio dataset is available upon request from its authors. The SIC dataset, a cropped hyperspectral image, can be provided upon request from the authors. All results are reproducible using the provided code, with no intermediate data stored.

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
