# Peer review of "Efficacy of Segmentation for Hyperspectral Target Detection"

_sensors, 2025, doi:10.3390/s25010272_

Round 1
Reviewer 1 Report
Comments and Suggestions for Authors
I would like to thank the authors for their efforts in producing this work. The paper reads well and brings something new to the field of research. However, i have some questions:
What previous studies have reported successful or unsuccessful segmentation results?
Are the resulting practical guidelines applicable to a variety of data sets, or are they limited to specific conditions?
The simulations based on synthetic are synthetic data, how are they generated, and do they faithfully reflect real-world scenarios?
How is the “representative algorithm” selected, and does it generalize well to other detection algorithms?
Reviewer 2 Report
Comments and Suggestions for Authors
1. The research motivation and main contributions should be included in the introduction.
2. The introduction should explain what problems exist in the current research and what issues this paper aims to address.
3. Section 3 should specify the location of data collection, camera resolution, lens focal length, acquisition frame rate, and weather conditions for each experiment.
4. The author should not only analyze the influencing factors of hyperspectral target detection but also propose methods to enhance the Segmentation Efficacy in Hyperspectral Target Detection.
5. Compare the method proposed by the author for enhancing the Segmentation Efficacy in Hyperspectral Target Detection with the methods in recent literature and experimentally validate the effectiveness of the author's proposed method.
